# Emerging Strategies Targeting Catabolic Muscle Stress Relief

**DOI:** 10.3390/ijms21134681

**Published:** 2020-06-30

**Authors:** Mattia Scalabrin, Volker Adams, Siegfried Labeit, T. Scott Bowen

**Affiliations:** 1School of Biomedical Sciences, University of Leeds, Leeds LS2 9JT, UK; M.Scalabrin@leeds.ac.uk; 2Department of Experimental and Molecular Cardiology, TU Dresden, Heart Center Dresden, 01307 Dresden, Germany; volker.adams@mailbox.tu-dresden.de; 3Dresden Cardiovascular Research Institute and Core Laboratories GmbH, 01067 Dresden, Germany; 4Medical Faculty Mannheim, University of Heidelberg, 68167 Mannheim, Germany; labeit@medma.de; 5Myomedix GmbH, Im Biengarten 36, 69151 Neckargemünd, Germany

**Keywords:** atrophy, diaphragm, disuse, mitochondria, MuRF1, MuRF2, skeletal muscle, ubiquitin, wasting

## Abstract

Skeletal muscle wasting represents a common trait in many conditions, including aging, cancer, heart failure, immobilization, and critical illness. Loss of muscle mass leads to impaired functional mobility and severely impedes the quality of life. At present, exercise training remains the only proven treatment for muscle atrophy, yet many patients are too ill, frail, bedridden, or neurologically impaired to perform physical exertion. The development of novel therapeutic strategies that can be applied to an in vivo context and attenuate secondary myopathies represents an unmet medical need. This review discusses recent progress in understanding the molecular pathways involved in regulating skeletal muscle wasting with a focus on pro-catabolic factors, in particular, the ubiquitin-proteasome system and its activating muscle-specific E3 ligase RING-finger protein 1 (MuRF1). Mechanistic progress has provided the opportunity to design experimental therapeutic concepts that may affect the ubiquitin-proteasome system and prevent subsequent muscle wasting, with novel advances made in regards to nutritional supplements, nuclear factor kappa-light-chain-enhancer of activated B cells (NFκB) inhibitors, myostatin antibodies, β_2_ adrenergic agonists, and small-molecules interfering with MuRF1, which all emerge as a novel in vivo treatment strategies for muscle wasting.

## 1. Introduction

Skeletal muscle is the largest and one of the most dynamic organs in the human body, representing 30–40% of total body mass and containing up to 75% of the organism’s protein reserves [1]. Skeletal muscle is essential for life, supporting movement, respiration, thermoregulation, and metabolic homeostasis. As such, skeletal muscle is a highly adaptable organ that can sense and respond accordingly to match environmental cues. Skeletal muscle wasting (also referred to as atrophy) is a characteristic of several catabolic conditions, including aging (i.e., sarcopenia), starvation, and immobilization, but also many acute and chronic illnesses such as cancer, heart failure, sepsis, and diabetes [2]. Cachexia is inextricably linked to muscle atrophy in many illnesses, however, the former is defined as a complex multifactorial metabolic syndrome that is associated with a significant reduction in body mass underpinned by skeletal muscle loss (with or without fat mass loss), which is not fully reversible with nutritional aids [3]. Collectively, therefore, a loss of muscle mass leads to a decline in functional mobility, which contributes to poor quality of life and survival [4]. Accordingly, our current lack of direct treatments to rescue muscle wasting across millions of patients is a key concern, with exercise training the only established intervention [5]. However, given that many patients are too ill or bedridden to perform exertional exercise, the development of novel pharmacological strategies to inhibit muscle atrophy represents an important research avenue. In this context, targeted inhibition of procatabolic factors specifically activated under wasting conditions could offer the most beneficial treatment strategy for patients [6,7,8].

The current review, therefore, will address novel approaches that may permit modulation of procatabolic factors upregulated across various wasting conditions that impact muscle homeostasis. We will first introduce a major mechanism in the myofiber responsible for degrading proteins and thus, atrophy: the ubiquitin-proteasome system (UPS) and its muscle activating muscle-specific E3 ligase Muscle RING-finger protein 1 (MuRF1). We then discuss recent advances in pharmacological aids that have shown potential in vivo application for inhibiting both upstream and downstream control nodes in this system and thus limit muscle wasting across various catabolic conditions.

## 2. Pathways Modulating Muscle Atrophy

Atrophy can be defined as a cellular shrinkage of tissues and organs caused by loss of organelles, cytoplasm, and proteins [2]. Skeletal muscle atrophy, therefore, is underpinned by complex changes in the balance between rates of protein synthesis and protein degradation, with many wasting conditions characterized by increased rates of proteolysis and suppressed rates of protein synthesis. As detailed extensively elsewhere [2], the molecular signaling pathways that interact to control protein synthesis and degradation generally include the insulin-like growth factor 1 (IGF1)/protein kinase B (Akt)/mammalian target of rapamycin (mTOR)-forkhead box protein O (FoxO), TGF-β/myostatin/bone morphogenetic protein (BMP), nuclear factor kappa-light-chain-enhancer of activated B cells (NFκB), and glucocorticoid. These signaling pathways can interact to determine the terminal fate of many proteins, which include degradation by one of the four major proteolytic systems in the cell: ubiquitin-proteasome, autophagy-lysosome, calpain, and caspase [9]. The signaling pathways and proteolytic systems controlling muscle mass are tightly governed by upstream factors related to hormonal/cytokine, metabolic/nutrient, mechanical load, and neural activity, which helps to explain why sudden changes to our environment such as starvation, exercise, immobilization, or disease result in rapid alterations to muscle mass. This review will primarily focus on UPS-dependent degradation, and we refer the reader to other reviews detailing the additional proteolytic systems [2,10,11].

### 2.1. Proteasome-Dependent Degradation

Characterized by a large protein complex (2.5 MDa) called the 26S proteasome, the most important proteolytic pathway recognized to mediate muscle atrophy is the UPS [12,13,14]. This enzyme complex degrades structural and regulatory proteins selectively targeted following the covalent attachment of polyubiquitin chains to lysine residues (Figure 1). The UPS is composed of an active 20S catalytic core capped by regulatory 19S subunits that are responsible for unfolding proteins for entry [15]. A family of three enzymes exert a major control on the UPS, termed E1, E2, and E3. As shown in Figure 1, the E1 activating enzyme initially primes the ubiquitin protein in an ATP-consuming reaction, while the E2 conjugating enzyme passes ubiquitin to the E3 ligase enzyme that links ubiquitin to the target protein in a repeated process (termed polyubiquitination) for subsequent recognition and degradation by the 26S proteasome. While proteasome-dependent degradation is considered a main fate for many sarcomeric proteins, some early events in this process include activated calpains and caspases (i.e., caspase-3 and -8) initially disrupting sarcomere assembly (by targeting Z-disks to release major proteins such as titin, nebulin, filamin, troponin-T, and desmin) as well as cleaving actomyosin to form a 14 kDa actin fragment for subsequent UPS processing [10,16,17]. One major rate-limiting step considered in the UPS is the attachment of ubiquitin to target proteins via specific E3 ubiquitin ligases, which have attracted particular attention given their potential for therapeutic modulation not only for muscle atrophy but also in the regulation of inflammation and immunity [18].

### 2.2. Muscle-Specific E3 Ligases: A Rate-Limiting Step in the UPS

A major breakthrough in the muscle field was achieved when multiple studies identified a series of so-called “atrogenes” transcriptionally upregulated in a variety of wasting conditions (i.e., diabetes, fasting, uremia, immobilization, and denervation) but otherwise not generally expressed under basal conditions [19,20]. Two key atrogenes identified were the muscle-specific E3 ubiquitin-ligases MuRF1 and muscle atrophy F-box (MAFBx; also termed atrogin-1) [19,20]. Evidence is now conclusive that knockdown of either gene can attenuate muscle wasting in various catabolic conditions including denervation, hindlimb suspension, fasting, chronic kidney disease, and lung disease [21], although their role in sarcopenia remains controversial [22].

In particular, removal of many thick filaments of the sarcomere is achieved by MuRF1-dependent targeting of troponin I, myosin heavy chain, myosin-binding protein C, myosin light chain [23,24] and actin [25]. In contrast, fewer MAFbx-dependent targets have been reported, and these have been specific to regulatory proteins linked to protein synthesis (e.g., MyoD and eukaryotic translation initiation factor 3 subunit F). In addition, other key E3 ligases that can accelerate sarcomeric proteolysis include: tripartite motif-containing protein 32 (Trim 32, reported to target thin filaments such as actin, tropomyosin, and troponins but also Z band-located proteins such as alpha-actinin and desmin) [26], tumor necrosis factor receptor-associated factor 6 (TRAF6, reported to target thick filaments and regulatory proteins) [27], C terminus of HSC70-Interacting Protein (CHIP, which targets filament C via lysosomal-dependent degradation) [28], F-Box Protein 40 (Fbxo40, targeting anabolic-insulin signaling proteins such as Insulin receptor substrate 1) [29], as well as muscle ubiquitin ligase of SCF complex in atrophy-1 (MUSA1) and specific of muscle atrophy and regulated by transcription (SMART), whose targets remain unclear [30]. The activation of E3 ligase transcription is highly regulated by numerous signaling pathways and transcription factors that include IGF1/Akt/FoxO, myostatin/mothers against decapentaplegic homolog (Smad) 2-3, NFκB, and glucocorticoid/glucocorticoid receptor. As such, these pathways have yielded useful therapeutic targets for inhibiting muscle wasting [2].

## 3. Therapeutic Treatments to Inhibit Muscle Atrophy

Currently, the only proven treatment for muscle wasting is exercise training [5], yet many patients are simply too ill, frail, or bedridden to perform physical exertion. Other non-physical exercise approaches that may also yield beneficial effects include certain nutritional supplements (e.g., amino acids) or hormonal therapy (e.g., testosterone) [7]. Given the key role that elevated proteolysis plays in mediating muscle atrophy across various conditions [21], the development of novel pharmacological aids for in vivo administration that target this process would represent a significant breakthrough. Recent advances in new methodologies such as proteomics, metabolomics, and transcriptomics have been instrumental in shedding new light on the underlying molecular pathways involved in regulating skeletal muscle wasting. However, inhibition of certain proteolytic pathways (via pharmacological or genetic deletion) have proven ineffective or even detrimental: suppressing autophagy impairs clearance of damaged organelles and misfolded proteins which leads to a dystrophic phenotype [31,32,33], while overexpression of calpastatin (an endogenous calpain inhibitor) in models of disuse-induced atrophy (hindlimb suspension) and sepsis has failed to impact muscle mass [34,35]. Similarly, previous studies have shown that exogenous inhibition of caspase is effective in preserving muscle function but not mass [36,37]. In contrast, much attention has been focused on targeting regulatory steps in proteasome-dependent degradation, with some encouraging treatments achieved by targeting both upstream (e.g., cytokines, transcription factors) and downstream (e.g., E3 ligases, 26S proteasome) nodes in the UPS (Figure 1).

### 3.1. Upstream Inhibition of UPS: Inflammatory Cytokines, Growth Factors, and Transcription Factors

During inflammatory diseases such as cancer, sepsis, or heart failure, an elevated and often chronic production of proinflammatory cytokines such as interleukin-1 (IL-1), interleukin-6 (IL-6), and tumor necrosis factor-alpha (TNFα) can act as upstream UPS triggers to induce atrophy [38]. Proinflammatory cytokines have been shown to modulate signaling via membrane-bound receptors activating a number of instrumental transcription factors that include NFκB, FoxO, and p38 MAPK, all of which are known to accelerate proteolysis involving MuRF1 (and MAFbx) activation [39,40,41,42]. Various studies have now shown that these effects can be inhibited by the use of novel antibodies (Figure 1). For example, the administration of MR16-1 antibody (anti-mouse IL-6 receptor antibody) in a mouse model of disuse-induced muscle atrophy (tail suspension) resulted in partial protection from muscle wasting alongside suppressed MuRF1 and MAFbx gene expression [42]. A similar study using Remicade^TM^ (an anti-TNFα drug that is clinically available) found that by interfering with TNFα activity, this drug reduced atrophy of dystrophic muscles by preventing NFκB translocation to the nucleus and, consequently, MuRF1 accumulation [40]. A link between cytokine-MuRF1 signaling has been confirmed in mice injected with exogenous TNFα [43], such that mice with genetic inactivation of MuRF1 were protected from force loss induced by TNFα treatment [43], which highlights an important role for MuRF1 in regulating myofibril contractile function as well as atrophy. In addition to cytokines, other secreted circulating factors known to induce muscle wasting include growth factors such as myostatin. Myostatin is a negative regulator of muscle mass and a member of the transforming growth factor-beta (TGF-β) superfamily, and acts by binding plasma membrane-located activin receptors type IIB and IIA (ActRIIB/A), which leads to subsequent phosphorylation of Smad2/3-Smad4 complex and inhibition of Akt signaling. An anti-myostatin antibody (ATA 842) administered for 4 weeks increased muscle mass in sarcopenic mice [44], while a soluble ActRIIB inhibitor rescued muscle wasting in cancer cachexia mice [45] (Figure 1).

Another member of the TGF-β superfamily, growth and differentiation factor 11 (GDF11), has been reported to work alongside myostatin to mediate muscle atrophy in vivo with MuRF1 involvement [46,47]. Treatment with GDF11 propeptide-Fc (a natural inhibitor of GDF11 derived from the GDF11 precursor protein) in dystrophic mdx mice (a model of Duchenne muscular dystrophy) improved muscle mass and strength but, to our knowledge, no data have been published on the effects of this treatment on MuRF1 expression [48]. Other approaches have targeted indirect myostatin inhibition by using small-molecule inhibitors (C188-9) to block the phosphotyrosyl peptide binding site and thus phosphorylation of Signal Transducer and Activator of Transcription 3 (Stat-3). Stat-3 is upregulated via IL-6-JAK signaling and can induce myostatin and MuRF1 expression via the transcription factor CCAAT-enhancer-binding protein delta (C/EBP-δ). Treatment with C188-9 suppressed MuRF1 expression alongside proteasome and caspase-3 activity, while preserving muscle mass in both cancer and diabetic mice [49].

Similar to myostatin and inflammatory cytokines, circulating glucocorticoids are also elevated under catabolic conditions and regulate atrophy via glucocorticoid receptor-dependent signaling that can increase MuRF1 expression (i.e., by direct or via FoxO signaling) [50]. However, treatment with the glucocorticoid receptor antagonist RU-38486 or RU 486 can suppress MuRF1, MAFbx, and proteasome levels in septic mice [51,52] (Figure 1). Since many catabolic conditions are associated with elevated oxidative stress that can directly promote proteasome-dependent degradation via MuRF1 and MAFBx expression, another approach has been to employ antioxidant treatments. Specifically, mitochondrial-targeted antioxidants have proven particularly effective in preventing muscle wasting in a range of conditions including mechanical ventilation, inflammation, denervation, and fasting [53,54,55] (Figure 1). However, given the important physiological and signaling roles played by reactive oxygen species (ROS), chronic blockade of these molecules could result in severe complications (e.g., cancer) [56].

An additional therapeutic target for muscle atrophy has been the selected inhibition of atrophic transcription factors such as NFκB and FoxO (Figure 1). Among these, the selective inhibitor IMD-0354 was developed in the late 90s to suppress NFκB [57]. However, despite promising results in vitro, administration of IMD-0354 during muscle unloading was unable to prevent MAFbx and MuRF1 accumulation as well as muscle wasting, despite preventing inhibitor of nuclear factor kappa-B kinase subunit beta (IκKβ) phosphorylation [58]. More recent experiments have been more encouraging, with a novel class of orally bioavailable NFκB inhibitors (termed edasalonexent and CAT-1041) able to inhibit muscle wasting in both murine and canine models of muscular dystrophy [46]. However, since multiple transcription factors control muscle mass, other studies have focused on inhibiting the FoxO family of transcription factors. Proteins belonging to this family of transcription factors, when not phosphorylated, translocate to the nucleus to increase MuRF1 and MAFbx transcription, and this can be controlled by Akt-dependent signaling [59,60]. Approaches targeting the Akt/FoxO signaling axis include administration of leucine [61,62] as well as its active metabolite Beta-Hydroxy b-methylbutyrate (HMB) [42,63,64,65,66]. Another candidate recently described in the literature is Matrine, an alkaloid found in the plant *Sophorae flavescentes* that showed strong antitumoral and anti-inflammatory activity [67]. This compound is approved by the China Food and Drug Administration for use in cachectic patients and was shown to attenuate MuRF1 mRNA expression and maintain fiber size via Akt/FoxO pathway in mice with cancer cachexia [67].

Another promising area has been the administration of the β_2_-adrenergic receptor (β_2_-AR) agonists, which can exert both pro-anabolic and anti-catabolic effects [68]. Conventional (e.g., formoterol) [69], as well as more novel β_2_-ARs such as 5-hydroxybenzothiazolone (5-HOB) [70] and espindolol/MT-102 [71,72], have shown benefits in promoting muscle growth and attenuating atrophy in experimental models of aging and cancer cachexia, possibly via NFκB/FoxO-dependent MuRF1 activation. However, the use of β_2_-AR can have adverse effects on cardiovascular function, which can have serious repercussions in many patients. Overall, while it seems that some viable treatments are available to inhibit multiple transcription factors and thus UPS activation, targeting a more central node where signaling networks converge, such as the ubiquitin-proteasome pathway *per se*, may be a more specific and thus beneficial approach.

### 3.2. Downstream Inhibition of UPS via the 26S Proteasome

As discussed earlier, muscle wasting often involves the degradation of polyubiquitinated proteins via the 26S proteasome [12]. Bortezomib (otherwise termed Velcade^TM^ or PS-341) is a selective boronic acid proteasome inhibitor approved by the United States Food and Drug Administration and used as a third-line treatment of multiple myeloma and mantle cell lymphoma [73]. Bortezomib functions by inhibiting the catalytic site of the proteasome complex without direct effects on ubiquitination or upstream activators [74]. Studies in murine models investigating the effects of bortezomib on muscle atrophy have produced mixed results showing either a significant reduction of muscle atrophy by up to 50% in the soleus muscle of denervated rats [75] or no effects in cancer mice [73]. Further experiments focused on the diaphragm have shown that bortezomib lowered proteasome activity and MAFBx/MuRF1 transcripts with normalized myosin protein levels and improved contractile function in heart failure rats [76], yet limited benefits were observed following acute mechanical ventilation-induced diaphragm atrophy [77,78]. Carfilzomib is a clinically approved irreversible selective proteasome inhibitor. Similar to bortezomib, this drug is employed as a second-line treatment for patients with multiple myeloma [79], with some evidence suggesting the efficacy of this drug to prevent muscle wasting and MuRF1 activity. For example, early treatment with Carfilzomib (2 mg/kg; 2 × per week) in mice with cancer-associated cachexia was effective in partly rescuing skeletal muscle wasting and, through the downregulation of angiotensin II, MuRF1 and MAFBx expression in skeletal muscle [80]. Other proteasome inhibitors tested include MG132, a reversible and cell-permeable proteasome inhibitor belonging to the class of synthetic peptide aldehydes. MG132 has been able to rescue muscle mass by ~50–75% alongside reducing the expression of both MuRF1 and MAFBx in mice following both limb immobilization [40,60] and cancer [81]. However, it is difficult to delineate the effects of MG132 on the proteasome per se, as this drug also inhibits the NFκB canonical pathway by preventing degradation of IκBα [60,81] as well as lysosomal proteases and calpains [40], with lack of clarity over benefits to muscle contractile function [82].

A major consideration for the treatment of proteasome inhibitors is that patients have shown dose-limiting toxicity, drug-resistance, and several adverse effects such as cardiac complications and even muscle weakness, which severely limit their application to the general population [26,83]. Overall, while proteasome-specific inhibitors have shown some benefits, there is a lack of consistency in positive outcomes, and it appears that maintaining proteasome-dependent degradation is essential for preserving cellular homeostasis [12]. As such, a more unique therapeutic approach that targets steps earlier in the UPS pathway, such as blocking the function of muscle-specific E3 ligases that are atrophy dependent, may be a more optimal approach with fewer side effects [6,11,26].

## 4. Targeted Small-Molecule Inhibition of the E3 Ligase MuRF1

There is a fast-growing field on how to target specific E3 ligases in different cellular contexts that were previously thought to be undruggable [84]. What evidence is there to support inhibiting one E3 ligase over another? Within the skeletal muscle context, there is good evidence favoring the MuRF1 protein over other E3 ligases. Based upon gene inactivation experiments, the deletion of MuRF1 is sufficient to provide partial protection from atrophy after denervation [19] and various other catabolic conditions [21]. In further support, MuRF1 is: (1) consistently upregulated in over 15 different settings of atrophy [21]; (2) central to catabolic rather than healthy basal conditions unlike other E3 ligases [85]; (3) able to regulate distinct pools of sarcomere-specific substrates that make up the majority of muscle bulk [86]; (4) muscle-specific which limits any adverse effects in other cell types [87,88,89]. As such, these data provide a strong rationale for developing effective in vivo drugs capable of suppressing MuRF1, but is this possible? In the next section, we discuss from a structural perspective the MuRF family and potential strategies for perturbing MuRF1 functions (i.e., properties that are specific to MuRF1 among the TRIM family) (Figure 2).

### 4.1. Structural Properties of the MuRFs

The muscle-specific RING finger proteins (MuRFs) are a group of three highly homologous proteins coded by different genes that are classified as part of the TRIpartite motif (TRIM) family, which include MuRF1 (TRIM 63) [90], MuRF2 (TRIM55) [91], and MuRF3 (TRIM64) [92,93]. MuRFs are expressed in skeletal and cardiac muscle and characterized by their RING finger, a unique MuRF-family-specific motif (MFC), a B-box, a coiled-coil (CC), and acidic-tail domains (Figure 2), which share high sequence identity apart from the N- and C-termini whose residue length differs considerably [92]. Like all TRIM family members, the RING motif is responsible for ubiquitination, B-box for protein binding, and CC domain for self-association.

As demonstrated in Figure 2, MuRF1′s RING domain was found in the N-terminal position with a conserved pattern of cysteine and histidine residues, which allowed attachment of ubiquitin to recognized substrates [94]. The central region, constituted by a B-box type II and a helical domain, is responsible for the interaction with different target proteins and is considered the functional unit of MuRF1 [94,95,96]. To our knowledge, while the full 3D structure of MuRF1 is yet to be reported, the B-box domain has been determined [97], which indicated MuRF1 self-associates and forms multimers. This self-multimerisation is presumably the reason why no structure of MuRF1 has been determined so far, as the structure of higher-order multimers would need to be resolved in the first place [97]. MuRF1 (and MuRF2) also presents a one signature-box (COS-box) sequence motif in its helical domain [95]. Evidence suggests that this domain plays a central role in protein recruitment by interacting with domains near the C-terminus of titin and so represents an interesting pharmacological target to limit protein activity [95,96,98]. In this regard, MuRF1 has received much attention for is its ability to bind at the M-line and interact with the large structural protein titin. Titin spans the entire sarcomere and is involved in regulating both sarcomeric structure and signaling [99]. While MuRF1 does not seem to target titin for degradation per se, this interaction may influence sarcomeric stability [100] and has recently been targeted by interfering small molecules to inhibit atrophy [82,101]. Beyond the M-line of the sarcomere, MuRF1 is also reported to locate at the Z-line, nucleus [102,103], and mitochondria [104]. While MuRF1 is best known for its involvement in sarcomeric protein degradation, more recently it has been recognized to have potential broad signaling effects that include regulation of protein synthesis, mitochondrial function, amino acid and carbohydrate metabolism, insulin homeostasis, apoptosis, and endoplasmic reticulum (ER) stress response [21,82,101,105] (Figure 3).

MuRF2 plays a central role in sarcomere assembly during development by transiently associating with titin and myosin [103]. Post-development, MuRF2 works in synergy with MuRF1 to mediate signal transduction in cardiomyocytes to prevent cardiac hypertrophy while stabilizing fast-fibers in skeletal muscle [82,103,106]. MuRF2 also interacts with the titin kinase in a stretch-dependent manner, regulating protein synthesis via the transcription factor serum response factor (SRF) [91]. Furthermore, recent evidence suggests that MuRF2 may act as an atrogene capable of regulating muscle wasting, as demonstrated in a mouse model of cardiac cachexia [106]. The final family member MuRF3 was the first of the MuRFs to be identified [93] and is required for skeletal myoblast differentiation, development of cellular microtubular networks and myogenesis [107,108,109]. Together with MuRF1, the MuRF3 protein localizes to the M-band and Z-disk, playing a key role in sarcomeric protein degradation [107]. MuRF3 KO mice have shown this ligase to be necessary for limiting cardiac abnormalities induced by heart failure [103] and diabetic cardiomyopathy [110]. Given similarities in sequence, structure, and functionality between MuRF1/2/3, some level of redundancy occurs, which has limited our understanding of the different roles played by each family member [108]. In addition, such close homology between MuRFs makes the development of inhibitors of this ligase family particularly challenging, with issues related to cross-reactivity [111] and potential lethality if multiple MuRFs are blocked simultaneously, as shown in mice with double gene knock-out of MuRF1/2 [112] or MuRF1/3 [108,109].

### 4.2. Developing Novel Small-Molecules to Inhibit MuRF1

Knowledge of MuRF1′s protein structure and comparison with other TRIM members suggests potential strategies for attenuating its function. These include interference with its coiled-coil region involved in protein recognition [112], its b-box domain involved in protein dimerization [92], or its MFC domain that is conserved between MuRFs but no other TRIMs [90]. Finally, direct targeting of the ubiquitin-transferring RING domain is a possible strategy. Another approach may be blocking MuRF1′s interaction with known targets that can influence protein turnover, such as the titin filament [113]. P013222 is a specific MuRF1 inhibitor developed over a decade ago, showing low toxicity [114]. In a cellular model of muscle atrophy (C2C12 cells treated with the synthetic glucocorticoid dexamethasone), this compound was able to inhibit MuRF1 autoubiquitination and, by stabilizing myosin heavy chain, to prevent its degradation in a dose-dependent manner (12.5–50 μM) [114]. Other approaches using adenoviral blockade to inhibit MuRF1 have shown similar results in myotubes stressed with dexamethasone [23,85]. To our knowledge, however, these inhibitors were never tested in vivo, thus limiting clinical translation and targeting inhibition of MuRF1′s catalytic RING domain, which seems necessary for maintaining muscle homeostasis [109]. Therefore, an alternative strategy might be to interfere with MuRF1’s recognition and binding domain towards targeted substrates, particularly those of the sarcomere, and leaving its RING finger unperturbed, as recently described [82,101]. This approach may have the advantage of maintaining the basal activity of MuRF1’s catalytic N-terminal RING domain while limiting possible risks of myopathy (e.g., genetic inactivation of MuRF1 and these can result in cardiac septum ruptures and accumulation of protein aggregates) [109].

One small molecule identified and characterized so far that targets MuRF1 coiled-coil region, as demonstrated in a series of multiple mouse models of muscle wasting, appears to have low toxicity, while functionally having the ability to interfere with MuRF1 target-recognition [82,101] (Figure 2). Intriguingly, in two mechanistically distinct murine models of cardiac cachexia, the administration of this small molecule mixed with chow (over 6–10 weeks) was able to reduce skeletal muscle MuRF1 expression and proteasome activity, while fiber contractile dysfunction and atrophy were attenuated [82,101]. We should stress that these studies are recent and still ongoing, and mostly derived from a single small molecule [82,101]. Clearly, they will require follow-up in different animal models and expanded toxicology. However, one of the most striking findings from these experiments using this specific small molecule was its ability to rescue force loss in stressed mice, which may be regulated by changes to both sarcomeric and metabolic proteins [82,101].

Expression of sarcomeric proteins that are important for force generation, such as actin and telethonin, were also normalized by treatment using this specific small molecule [82,101]. We have also started exploratory proteomics studies to identify potential underlying pathways affected by this identified small molecule that may regulate muscle function. This approach has revealed that proteins participating in protein synthesis, apoptosis, and mitochondrial ATP synthesis were protected under stress when treated with this specific small molecule [82,101]. Consistent with this, we have found treatment with this small molecule can partially rescue muscle mitochondrial morphology and function (e.g., porin content, citrate synthase activity, mitochondrial complex I activity, and mitochondrial ribosomal Protein S5) [82], which are possibly in line with earlier yeast two-hybrid studies that showed that MuRF1 can preferentially target mitochondrial in addition to sarcomeric proteins [115], and implicate possible localization within mitochondria as recently reported in cardiac muscle [116]. Furthermore, genetically engineered mice have shown that MuRF1 can negatively regulate pyruvate dehydrogenase (PDH) function, insulin sensitivity, and hepatic glycogen stores [117], raising the intriguing proposition that MuRF1 could play a role in driving insulin-resistance associated with muscle wasting [118]. Overall, these data may support previous evidence that MuRF1 possesses broad signaling effects within both the myofiber and whole-organism (Figure 3).

It should be noted that this identified small molecule can also affect MuRF2 protein expression [82]. The construct used for screening had high homology between MuRF1 and MuRF2 since both MuRFs share very high homology in their MFC-Bbox-cc segments (Figure 2). Also, the genetic knockdown of MuRF1 has been shown to affect MuRF2 [106,119]. Finally, recent experiments have confirmed that mice with a genetic deletion of MuRF2 do not suffer muscle wasting in cardiac cachexia [106]. Considering the central coiled-coil fragment was used to develop the inhibitor, and this domain remains highly conserved between MuRF1 and MuRF2, further in vivo and in vitro experiments are warranted to clarify the specificity of this identified small molecule [82,101]. In addition, future work is required to determine the metabolism, bioavailability, and pharmacokinetics of this small molecule as well as optimal treatment durations, doses, and potential adverse effects. While the development of simultaneous MuRF1/2/3 inhibitors remains an intriguing possibility, careful consideration is required as parallel genetic knockdown of multiple MuRFs can induce severe cardiac failure despite skeletal muscle hypertrophy [112,116].

### 4.3. Other Novel Approaches to Inhibit MuRF1 Function

Rather than direct inhibition, other emerging studies have touted alternative approaches to target MuRF1 blockade. For example, MuRF1 was recently shown to be post-translationally regulated by attachment of small ubiquitin-like modifier (SUMO)-1 via the E2/E3 ligases UBC9 and PIASγ/4 at the coiled-coil domain [120], which may represent another pharmacological target to modulate MuRF1. Another ubiquitin-like protein is neuronal precursor cell-expressed developmentally downregulated protein 8 (NEDD8). This protein shares a higher amino acid homology (60%) with ubiquitin compared to SUMO (about 20%) and is ubiquitously expressed in all cell types with the highest expression observed in the heart and skeletal muscle [121]. Under catabolic conditions, NEDD8 has been proposed to function as a Ub substitute in order to prevent depletion of the ubiquitin pool, with recent proteome-wide high-throughput screens showing NEDD8 is a MuRF1 interacting partner [122]. Treatment of C2C12 cells exposed to the glucocorticoid dexamethasone with the neddylation inhibitor MLN4924 has shown promising effects, limiting myosin heavy chain loss by suppressing MuRF1 expression [122]. As such, further studies are clearly warranted to fully assess the efficacy of MLN4924 and the impact that NEDD8 inhibition may have to rescue skeletal muscle function and mass in vivo. Furthermore, PROteolysis TArgeting Chimeras (PROTACs) is a promising and appealing technology able to regulate protein function by degrading target proteins instead of inhibiting them. More specifically, these hetero-bifunctional molecules recruit an E3 ubiquitin ligase to a given substrate protein resulting in its targeted degradation [123]. This approach, that resembles a chemical knockdown strategy, shows more sensitivity to drug-resistant targets and a greater chance to regulate non-enzymatic functions [124]. As such, implementing this technology to target specific interactions of MuRF1 with known targets could be one promising strategy to combat muscle wasting. In addition, others have identified novel MuRF1-E2 networks [125], suggesting that targeting the E2 ligases interacting with MuRF1 could offer a new therapeutic opportunity towards inhibiting atrophy.

In contrast, other groups have also focused on small-molecule therapeutics using non-biased approaches without any predefined targets based upon mRNA signatures in atrophy [126]. These studies have resulted in anabolic signaling pathways being primarily modulated as a mean to inhibit subsequent proteolysis, with ursolic acid and tomatidine identified as novel small-molecules that suppress MuRF1 expression and muscle wasting in aging, starvation and disuse atrophy, by blocking expression of activating transcription factor 4 (ATF4)-Gadd45a/MEKK4 kinase complex activation [126]. While these experimental developments for MuRF1 inhibition are exciting, we should note that other approaches in the past have been able to reduce MuRF1 levels in line with improved muscle mass and function. These include, for example, nutritional supplements (e.g., leucine; see Figure 1) [61,62,98] as well as aerobic exercise training [127,128].

## 5. Conclusions

Skeletal muscle wasting represents a common trait in aging and many clinical settings, including cancer, heart failure, and critical illness, characterized by a shift towards a procatabolic state. During the past few years, an increasing number of experimental approaches have succeeded to attenuate muscle wasting in animal models, which have targeted procatabolic factors as a new emerging paradigm to support muscle function. In particular, strategies that have been able to suppress the activity of the main cellular pathway responsible for proteolysis (i.e., the UPS), and the upstream E3 ligase MuRF1 using small-molecules, have proven highly beneficial in the attenuation of muscle atrophy. However, an optimal approach to reduce muscle atrophy is likely via the combination of various therapies tailored to the patient’s current condition, which could include exercise training, leucine supplementation, β_2_-ARs, and small molecule inhibitors of MuRF1. This holistic approach is probably the most powerful stimuli for inhibiting a variety of procatabolic factors and thus muscle atrophy. Next, large animal models and ultimately, human patients will be required to validate many of these novel experimental concepts.

## Figures and Tables

**Figure 1 ijms-21-04681-f001:**
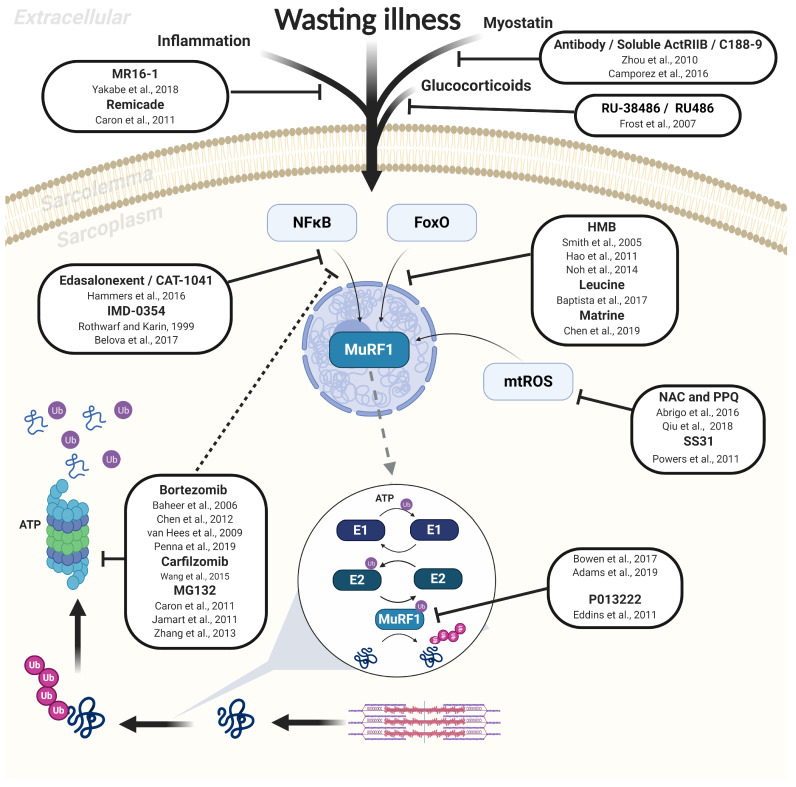
Overview of proteasome-dependent protein degradation via the muscle-specific E3 ligase MuRF1 as induced in various wasting illness, with recognized pharmacological inhibitors also presented. The most important proteolytic pathway to mediate muscle atrophy is the ATP-dependent ubiquitin-proteasome system, whereby structural and regulatory proteins are degraded by the 26S proteasome following polyubiquitination that involves activation of E1, E2, and E3 enzymes. MuRF1 (an E3 ligase) plays a fundamental role in tagging proteins with ubiquitin during catabolic conditions, which is activated by various stimuli and signaling pathways that include NFκB and FoxO. A selection of reported pharmacological inhibitors of MuRF1-dependent muscle atrophy are presented along with the study reference. Solid lines indicate a direct effect on the target, while dotted lines indicate a secondary indirect effect. See the main text for full details.

**Figure 2 ijms-21-04681-f002:**
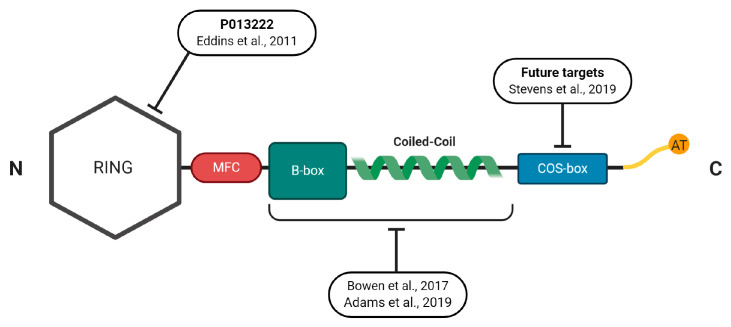
Schematic representation of MuRF1′s protein structure indicating the regions that have been, or could be, exploited for small molecule inhibition alongside the relevant study reference. See the main text for full details.

**Figure 3 ijms-21-04681-f003:**
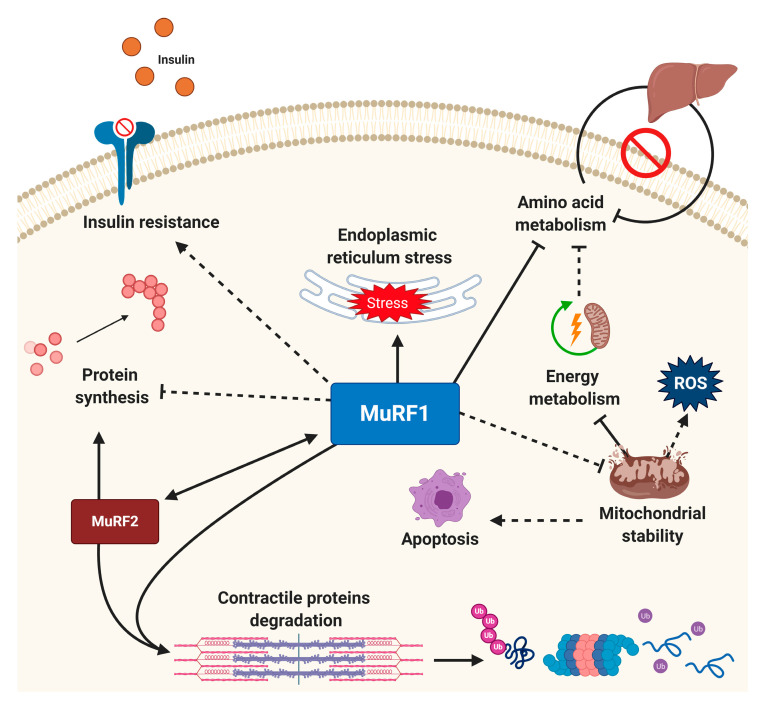
MuRF1 as a potential central node for regulating cellular homeostasis. While MuRF1 is known to play a primary role in sarcomeric protein degradation, recent evidence indicates it may govern multiple cellular processes that include mitochondrial health, apoptosis, insulin resistance, endoplasmic reticulum stress response, amino acids, and carbohydrate metabolism. MuRF1 also works in synergy with MuRF2 to mediate muscle wasting. Solid lines indicate mechanisms with strong supporting evidence for direct effects MuRF1 effects, with recently emerging mechanisms that require further validation represented by dotted lines (direct or indirect). See the main text for full details.

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
