# Peer review of "Emerging Strategies Targeting Catabolic Muscle Stress Relief"

_ijms, 2020, doi:10.3390/ijms21134681_

Round 1

Reviewer 1 Report

The present review discusses novel approaches that may allow modulation of pro-catabolic factors upregulated under various wasting conditions that affect skeletal muscle. This is an excellent review, with well-organized logical structure and clear statements. The authors sequentially consider the pathways modulating muscle atrophy and then primarily focus on the ubiquitin proteasome system (UPS)-dependent degradation as the most important proteolytic pathway recognized to mediate muscle wasting.

The main strength of this review is in complete, thorough analysis of the known therapeutic treatments to inhibit muscle atrophy. The authors describe in detail the approaches based on inhibition of upstream regulators (e.g. atrophic transcription factors) and downstream components (the 26S proteasome) of the ubiquitin proteasome system. Finally, targeted small-molecule inhibition of the UPS-activating muscle-specific E3 ligase MuRF1 is discussed.

The review contains an extensive analysis of the literature, is well written and absolutely clear. I see no major flaws. In summary, it is certainly very interesting and, in my opinion, should be of great interest to the readers.

Reviewer 2 Report

Authors present a quality and well-written review that describes emerging strategies targeting catabolic muscle stress relief with focus on MuRF1 E3 ligase.

Authors discuss recent progress in understanding the molecular pathways involved in regulating skeletal muscle wasting with a focus on pro-catabolic factors, in particular the ubiquitin proteasome system and its activating muscle-specific E3 ligase MuRF1.

Comments:

- Figure 1 and Section 3.2 = please add information about Carfilzomib, a clinically approved 26S Proteasome inhibitor

- Authors are encouraged to add chemical structures of the compounds wherever appropriate (especially, Fig1 and sections 3.2, 4.2)

- please consider adding a brief information about potential use of MuRF1 for development of  PROTACs

- If there is any X-ray data on MuRF1 ligase (i.e. in PDB there 1-2 of them available) = please add a figure to demonstrate tertiary structure of the protein and potential druggable pockets

- is NEDD8 (similar to SUMO) somehow involved in regulation of MuRF1 ? If yes = then please add information about its small molecule inhibitor MLN4924 as an additional regulator of MuRF1 pathway

- Please add a brief information about potential proteasome-independent role of MuRF1

- Conclusion is clearly missing elaboration on UPS/Proteasome aspect and therapeutic potential of targeting MuRF1 using small molecules

- Authors are kindly requested to cite the following article (DOI 10.3389/fphar.2018.00450) that describes RING-type E3 ubiquitin ligases as promising new therapeutic targets for regulation of inflammation and immunity

Reviewer 3 Report

This review discusses recent progress in regulating skeletal muscle wasting. This is an interesting and important paper focus on UPS-dependent degradation. These authors provided several approaches to demonstrate their conclusion.

No further comments.

Author Response

Thank you for the positive comments. 

Round 2

Reviewer 2 Report

The authors have addressed all the points. Manuscript has been improved significantly and can be accepted for publication.